# A Catalyst Framework for Minimax Optimization

**Junchi Yang**
UIUC
junchiy2@illinois.edu

**Siqi Zhang**
UIUC
siqiz4@illinois.edu

**Negar Kiyavash**
EPFL
negar.kiyavash@epfl.ch

**Niao He**
UIUC & ETH Zurich
niao.he@inf.ethz.ch

## Abstract

We introduce a generic *two-loop* scheme for smooth minimax optimization with strongly-convex-concave objectives. Our approach applies the accelerated proximal point framework (or Catalyst) to the associated *dual problem* and takes full advantage of existing gradient-based algorithms to solve a sequence of well-balanced strongly-convex-strongly-concave minimax problems. Despite its simplicity, this leads to a family of near-optimal algorithms with improved complexity over all existing methods designed for strongly-convex-concave minimax problems. Additionally, we obtain the first variance-reduced algorithms for this class of minimax problems with finite-sum structure and establish faster convergence rate than batch algorithms. Furthermore, when extended to the nonconvex-concave minimax optimization, our algorithm again achieves the state-of-the-art complexity for finding a stationary point. We carry out several numerical experiments showcasing the superiority of the Catalyst framework in practice.

## 1   Introduction

Minimax optimization has been extensively studied in past decades in the communities of mathematics, economics, and operations research. Recent years have witnessed a surge of its applications in machine learning, including generative adversarial networks [16], adversarial training [47, 28], distributionally robust optimization [31, 1], reinforcement learning [8, 9], and many others. The problem of interest in such applications is often a smooth minimax optimization problem (also referred to as saddle point problems):

$$\min_{x \in \mathcal{X}} \max_{y \in \mathcal{Y}} f(x, y), \tag{1}$$

where the function $f : \mathbb{R}^{d_1} \times \mathbb{R}^{d_2} \to \mathbb{R}$ is smooth (i.e., gradient Lipschitz), $\mathcal{X}$ is a convex set in $\mathbb{R}^m$, and $\mathcal{Y}$ is a convex and compact set in $\mathbb{R}^n$. In many machine learning applications, $f$ has a finite sum structure, that is $f(x, y) = \frac{1}{n} \sum_{i=1}^{n} f_i(x, y)$, where each component corresponds to a loss associated with single observation.

A significant body of first-order algorithms for minimax optimization exists in the literature, ranging from the classical projection method [42], Korpelevich's extragradient method [20], Nemirovski's Mirror Prox algorithm [32], Nesterov's dual extrapolation method [34], Tseng's accelerated proximal gradient algorithm [46], to many recent hybrid or randomized algorithms, e.g., [30, 17, 38, 19, 6, 25], just to name a few. Most of these existing work and theoretical analyses are limited to the following settings (i) the strongly-convex-strongly-concave setting (e.g., [45, 29, 15]), (ii) the general convex-concave setting (e.g., [32, 34]), and (iii) the special bilinear convex-concave setting (e.g., [5, 48, 7]). The lower complexity bounds for these three settings established in [50], [33], [37], respectively,

Table 1: Comparison with other algorithms for general strongly-convex-concave setting. For simplicity, we ignore the dependency on $\ell$ and $\mu$ inside $\log$. The lower complexity bound is $\Omega\left(\ell/\sqrt{\mu\epsilon}\right)$ [37]. Last column indicates whether the algorithm requires the knowledge of target accuracy $\epsilon$ in advance. All algorithms here have the $\tilde{\mathcal{O}}(\mathcal{D}_{\mathcal{Y}})$ dependency on the diameter of $\mathcal{Y}$ in oracle complexity.

| Algorithms | Oracle Complexity | number of loops | prefix $\epsilon$ |
|---|---|---|---|
| MINIMAX-APPA [24] | $\mathcal{O}\left(\ell/\sqrt{\mu\epsilon}\log^3(1/\epsilon)\right)$ | 3 | Yes |
| DIAG [44] | $\mathcal{O}\left(\ell^{\frac{3}{2}}/(\mu\sqrt{\epsilon})\log^2(1/\epsilon)\right)$ | 3 | No |
| Primal-dual Smoothing [51] | $\mathcal{O}\left(\ell^{\frac{3}{2}}/(\mu\sqrt{\epsilon})\log^2(1/\epsilon)\right)$ | 3 | Yes |
| Our Algorithm | $\mathcal{O}\left(\ell/\sqrt{\mu\epsilon}\log(1/\epsilon)\right)$ | 2 | No |

can be attained by some existing algorithms. For example, extragradient method (EG) achieves the optimal $\mathcal{O}(1/\epsilon)$ complexity for smooth convex-concave minimax problems, and the optimal $\mathcal{O}(\kappa\log(1/\epsilon))$ complexity for *well-balanced* strongly-convex-strongly-concave minimax problems, where the $x$-component and $y$-component of the objective share the same condition number $\kappa$ [50].

However, there are relatively few results outside of these settings. Of particular interests are the following two settings: $f(x, \cdot)$ is concave but not strongly-concave for any $x \in \mathcal{X}$, while $f(\cdot, y)$ could be strongly-convex or even nonconvex. Strongly-convex-concave minimax optimization covers broad applications in game theory, imaging, distributionally robust optimization, etc. While the special bilinear case of this setting has been studied extensively in the literature, the general case is less explored. In fact, strongly-convex-concave minimax optimization has also been routinely used as a building block for solving nonconvex-concave minimax problems [40, 44]. Hence, we mainly focus on the strongly-convex-concave setting.

For strongly-convex-concave minimax problems, the lower complexity bound of first-order algorithms is $\Omega\left(\ell/\sqrt{\mu\epsilon}\right)$ for achieving an $\epsilon$-duality-gap [37], where $\ell$ is the smoothness constant and $\mu$ is the strong convexity constant. Recently, [44] proposed the so-called dual implicit accelerated gradient algorithm (DIAG) that achieves the first-order oracle complexity of $\mathcal{O}\left(\ell^{3/2}/(\mu\sqrt{\epsilon})\log^2(1/\epsilon)\right)$. A similar complexity bound was obtained from the primal-dual smoothing method in [51]. More recently, [24] introduced the MINIMAX-APPA algorithm that further improves the complexity by shaving off a factor of $\mathcal{O}(\sqrt{\ell/\mu})$, yielding a near-optimal convergence rate up to the logarithmic factor. However, these algorithms are fairly complicated as they stack several procedures including accelerated gradient descent on $x$, accelerated gradient ascent on $y$, and accelerated proximal point algorithm, in different manners, thus requiring at least three loops. In addition to the complicated procedure, the latter two algorithms require an additional layer of smoothing, and solve the surrogate problem $\min_{x \in \mathcal{X}} \max_{y \in \mathcal{Y}} f(x, y) + \mathcal{O}(\epsilon)\|y\|^2$. In practice, how to select a good smoothing parameter of order $\mathcal{O}(\epsilon)$ remains elusive.

Meanwhile, it is unclear how these sophisticated algorithms can be integrated with variance-reduction techniques to solve strongly-convex-concave minimax problems with finite-sum structure efficiently. Most existing variance-reduced algorithms in minimax optimization focus on strongly-convex-strongly-concave setting, e.g., SVRG and SAGA [38], SPD1-VR [43], SVRE [6], Point-SAGA [26], primal-dual SVRG [11], variance reduced prox-method [4], etc. These algorithms typically preserve the linear convergence of batch algorithms, yet with cheaper per-iteration cost and improved complexity. Outside of this regime, few results are known [27, 49]. To the best of our knowledge, the design of efficient variance reduction methods for finite-sum structured minimax problems under the strongly-convex-concave or nonconvex-concave settings remains largely unexplored.

This raises the question: *can we simply leverage the rich off-the-shelf methods designed for strongly-convex-strongly-concave minimax problems to these unexplored settings of interest?* Inspired by the success of the Catalyst framework and accelerated APPA that use gradient-based algorithms originally designed for strongly convex minimization problems to minimize convex/nonconvex objectives [22, 21, 39, 13], we introduce a generic Catalyst framework for minimax optimization. Rooted in an inexact accelerated proximal point framework, the idea is to repeatedly solve the following auxiliary strongly-convex-strongly-concave problem using an existing method $\mathcal{M}$:

$$\min_{x \in \mathcal{X}} \max_{y \in \mathcal{Y}} f(x, y) + \frac{\tau_x}{2}\|x - \bar{x}_t\|^2 - \frac{\tau_y}{2}\|y - z_t\|^2. \tag{2}$$

While the algorithmic extension looks straightforward, selecting appropriate proximal parameters $\tau_x$, $\tau_y$, the prox centers $\bar{x}_t$, $z_t$, and the method $\mathcal{M}$ for solving the auxiliary problems, are critical and make a huge difference in the overall complexity. Our key insight is that when the condition numbers of the auxiliary problems are well balanced, they become relatively easy to solve and simply applying existing algorithms such as extragradient method as $\mathcal{M}$ would suffice. For instance, in the strongly-convex-concave setting, we set $\tau_x = 0, \tau_y = \mu$. In sharp contrast, the MINIMAX-APPA algorithm [24] uses $\tau_x = \frac{1}{\ell}$ and $\tau_y = \mathcal{O}(\epsilon)$, which results in extra complications (i.e., a two-loop algorithm) in solving the auxiliary problems.

Based on the generic Catalyst framework, we establish a number of interesting results:

(i) For strongly-convex-concave minimax optimization, we develop a family of two-loop algorithms with near-optimal complexity and reduced order of the logarithmic factor. In fact, simply combing Catalyst with extragradient method yields the complexity, $\mathcal{O}\left(\ell/\sqrt{\mu\epsilon}\log(1/\epsilon)\right)$, which improves over all existing methods, as shown in Table 1.

(ii) For nonconvex-concave minimax optimization, we provide a simple two-time-scale inexact proximal point algorithm for finding an $\epsilon$-stationary point that matches the state-of-the-art complexity of $\tilde{\mathcal{O}}\left(\ell^2\epsilon^{-3}\right)$.

(iii) For minimax problems with finite-sum structure, we provide a family of variance-reduced algorithms for the strongly-convex-concave setting, improving the $\tilde{\mathcal{O}}\left(n\bar{\ell}/\sqrt{\mu\epsilon}\right)$ complexity of the best batch algorithm to $\tilde{\mathcal{O}}\left(\bar{\ell}^2/\sqrt{\mu^3\epsilon}\vee n^{\frac{3}{4}}\bar{\ell}^{\frac{1}{2}}/\sqrt{\epsilon}\right)$, and to $\tilde{\mathcal{O}}\left(\bar{\ell}/\sqrt{\mu\epsilon}\vee n^{\frac{1}{2}}\bar{\ell}^{\frac{1}{2}}/\sqrt{\epsilon}\right)$ with additional assumption on cocoercive gradient. When extending to the nonconvex-concave setting, we improve the $\tilde{\mathcal{O}}\left(n\bar{\ell}^2\epsilon^{-3}\right)$ complexity of the best batch algorithm to $\tilde{\mathcal{O}}\left(n^{\frac{3}{4}}\bar{\ell}^2\epsilon^{-3}\right)$, and to $\tilde{\mathcal{O}}\left(n^{\frac{1}{2}}\bar{\ell}^2\epsilon^{-3}\right)$ with cocoercive gradient. Here $\bar{\ell}$ is the average of smoothness constants of the components.

For the ease of notation, we refer to the strongly-convex-strongly-concave setting as SC-SC for short, or $(\mu_1, \mu_2)$-SC-SC if the strong convexity and strong concavity constants are given by $\mu_1, \mu_2$. Similarly, SC-C or $\mu$-SC-C refers to the strongly-convex-concave setting, and NC-C to the nonconvex-concave setting. Throughout the paper, $\|\cdot\|$ stands for the standard $\ell_2$-norm.

## 2 A Catalyst Framework for SC-C Minimax Optimization

In this section, we focus on solving strongly-convex-concave minimax problems and introduce a general Catalyst scheme. We formally make the following assumptions.

**Assumption 1** (SC-C). *$f(\cdot, y)$ is $\mu$-strongly-convex for any $y$ in $\mathcal{Y}$, i.e.,*

$$f(x_1, y) \geq f(x_2, y) + \nabla_x f(x_2, y)^T(x_1 - x_2) + \frac{\mu}{2}\|x_1 - x_2\|^2, \quad \forall x_1, x_2 \in \mathcal{X}.$$

*and $f(x, \cdot)$ is concave for all $x$ in $\mathcal{X}$. $\mathcal{X}$ and $\mathcal{Y}$ are convex and closed sets, and $\mathcal{Y}$ is bounded with diameter $\mathcal{D}_\mathcal{Y} = \max_{y, y' \in \mathcal{Y}}\|y - y'\|$. There exists at least one saddle point $(x^*, y^*) \in \mathcal{X} \times \mathcal{Y}$, which satisfies $\max_{y \in \mathcal{Y}} f(x^*, y) \leq f(x^*, y^*) \leq \min_{x \in \mathcal{X}} f(x, y^*)$ for all $(x, y) \in \mathcal{X} \times \mathcal{Y}$.*

**Assumption 2** (Lipschitz gradient). *There exists a positive constant $\ell$ such that*

$$\max\{\|\nabla_y f(x_1, y_1) - \nabla_y f(x_2, y_2)\|, \|\nabla_x f(x_1, y_1) - \nabla_x f(x_2, y_2)\|\} \leq \ell[\|x_1 - x_2\| + \|y_1 - y_2\|],$$

*holds for all $x_1, x_2 \in \mathcal{X}, y_1, y_2 \in \mathcal{Y}$.*

The goal is to find an $\epsilon$-saddle point $(\bar{x}, \bar{y})$ such that $\text{gap}_f(\bar{x}, \bar{y}) := \max_{y \in \mathcal{Y}} f(\bar{x}, y) - \min_{x \in \mathcal{X}} f(x, \bar{y}) \leq \epsilon$. We call $\text{gap}_f(\bar{x}, \bar{y})$ the primal-dual gap, which implies both primal optimality gap and dual optimality gap. If $\epsilon = 0$, then $(\bar{x}, \bar{y})$ is a saddle point.

We present a generic Catalyst scheme in Algorithm 1. Analogous to its prototype [22, 39], this scheme consists of several important components: an inexact accelerated proximal point step as the wrapper, a linearly-convergent first-order method $\mathcal{M}$ as the workhorse, as well as carefully chosen parameters and stopping criteria.

---
**Algorithm 1** `Catalyst for SC-C Minimax Optimization`
---
1: Input: initial point $(x_0, y_0)$, parameter $\tau > 0$
2: Initialization: $\alpha_1 = 1, v_0 = y_0$
3: **for all** $t = 1, 2, ..., T$ **do**
4:     Set $z_t = \alpha_t v_{t-1} + (1 - \alpha_t) y_{t-1}$.
5:     Find an inexact solution $(x_t, y_t)$ to the following problem with algorithm $\mathcal{M}$

$$\min_{x \in \mathcal{X}} \max_{y \in \mathcal{Y}} \left[ \tilde{f}_t(x, y) := f(x, y) - \frac{\tau}{2} \|y - z_t\|^2 \right] \tag{$\star$}$$

   such that
$$f(x_t, y_t) - \min_{x \in \mathcal{X}} f(x, y_t) \leq \epsilon^{(t)} \text{ and } \nabla_y \tilde{f}_t(x_t, y_t)^T (y - y_t) \leq \epsilon^{(t)}, \forall y \in \mathcal{Y} \tag{3}$$

6:     $v_t = y_{t-1} + \frac{1}{\alpha_t}(y_t - y_{t-1})$;
7:     Choose $\alpha_{t+1} \in [0, 1]$ such that $\frac{1 - \alpha_{t+1}}{\alpha_{t+1}^2} = \frac{1}{\alpha_t^2}$.
8: **end for**
9: Output: $(\bar{x}_T, y_T)$ with $\bar{x}_T = \sum_{t=1}^T \frac{1/\alpha_t}{\sum_{m=1}^T 1/\alpha_m} x_t$.
---

**Inexact accelerated proximal point step.** The main idea is to repeatedly solve a series of regularized problems by adding a quadratic term in $y$ to the original problem:

$$\min_{x \in \mathcal{X}} \max_{y \in \mathcal{Y}} \left[ \tilde{f}_t(x, y) := f(x, y) - \frac{\tau}{2} \|y - z_t\|^2 \right], \tag{$\star$}$$

where $\tau > 0$ is a regularization parameter (to be specified later) and $z_t$ is the prox-center. The prox-centers $\{z_t\}_t$ are built on extrapolation steps of Nesterov [35]. Noticeably, this step can also be viewed as applying the original Catalyst scheme [22] to the dual function $h(y) := \min_{x \in \mathcal{X}} f(x, y)$. The major distinction is that we do not have access to the closed-form dual function, which causes difficulty in measuring the inexactness of auxiliary problems and evaluating the solution performance in terms of the primal-dual gap, instead of dual optimality.

**Linearly-convergent algorithm $\mathcal{M}$.** By construction, the series of auxiliary problems $(\star)$ are $(\mu, \tau)$-SC-SC. Thus, they can be solved by a wide spectrum of first-order algorithms established in the literature, at a linear convergence rate, including gradient descent ascent (GDA), extra-gradient method (EG), optimistic gradient descent ascent (OGDA), SVRG, to name a few. Yet, the dependence on the condition number may vary across different algorithms. We assume that any deterministic algorithm $\mathcal{M}$ when solving the $(\mu, \tau)$-SC-SC minimax problem has a linear convergence rate such that

$$\|x_k - x^*\|^2 + \|y_k - y^*\|^2 \leq \left(1 - \frac{1}{\Delta_{\mathcal{M},\tau}}\right)^k [\|x_0 - x^*\|^2 + \|y_0 - y^*\|^2], \tag{4}$$

and any stochastic algorithm $\mathcal{M}$ satisfies

$$\mathbb{E}[\|x_k - x^*\|^2 + \|y_k - y^*\|^2] \leq \left(1 - \frac{1}{\Delta_{\mathcal{M},\tau}}\right)^k [\|x_0 - x^*\|^2 + \|y_0 - y^*\|^2], \tag{5}$$

where $\Delta_{\mathcal{M},\tau}$ depends on $\tau$ and algorithm $\mathcal{M}$. For instance, when EG or OGDA is adopted, $\Delta_{\mathcal{M},\tau} = \frac{\ell + \tau}{4\min\{\mu, \tau\}}$ [45, 15, 2]; when SVRG or SAGA is adopted, $\Delta_{\mathcal{M},\tau} \propto n + \left(\frac{\ell + \tau}{\min\{\mu, \tau\}}\right)^2$, provided that the objective has the finite-sum structure and each component is $\ell$-smooth [38].

**Stopping criteria.** To guarantee the overall convergence in terms of primal-dual gap, it is necessary to approximately solve the auxiliary problem $(\star)$ to moderate accuracy and ensure the entire pair $(x, y)$ converges properly. For the sake of generalization, we adopt the criterion specified in (3) in our generic scheme. The stopping criterion can be achieved by most existing minimax optimization algorithms after sufficient iterations. Yet, it could still be hard to check in practice because $\min_{x \in \mathcal{X}} f(x, y_t)$ and $\max_{y \in \mathcal{Y}} \nabla_y \tilde{f}_t(x_t, y_t)^T (y - y_t)$ are not always computable. The following lemma shows that this issue can be alleviated, at the minor cost of a full gradient evaluation and a projection step.

**Lemma 2.1.** *Consider a function $\tilde{f}(x, y)$ that is $(\mu_1, \mu_2)$-SC-SC and has $\tilde{\ell}$-Lipschitz gradient on $\mathcal{X} \times \mathcal{Y}$. Let $z^* = (x^*, y^*)$ be the saddle point, i.e, the solution to the minimax optimization*

$\min_{x \in \mathcal{X}} \max_{y \in \mathcal{Y}} \tilde{f}(x, y)$. *For any point $z = (x, y)$ in $\mathcal{X} \times \mathcal{Y}$, we define $[z]_\beta = ([x]_\beta, [y]_\beta)$ with $\beta > 2\tilde{\ell}$ to be the point after one step of projected gradient descent ascent:*

$$[x]_\beta = \mathcal{P}_{\mathcal{X}}\left(x - \tfrac{1}{\beta}\nabla_x \tilde{f}(x, y)\right), \quad [y]_\beta = \mathcal{P}_{\mathcal{Y}}\left(y + \tfrac{1}{\beta}\nabla_y \tilde{f}(x, y)\right),$$

*then we have*

1. $\operatorname{gap}_{\tilde{f}}([z]_\beta) \leq A\|z - z^*\|^2, \quad \nabla \tilde{f}([x]_\beta, [y]_\beta)^T(\bar{y} - [y]_\beta) \leq A\|z - z^*\|^2 + 2\beta\mathcal{D}_{\mathcal{Y}}\|z - z^*\|$;

2. $\|z - z^*\| \leq \frac{\beta + \tilde{\ell}}{\tilde{\mu}}\|z - [z]_\beta\|, \quad \|z - [z]_\beta\|^2 \leq \frac{2}{(1 - \tilde{\ell}/\beta)^3}\|z - z^*\|^2,$

*where $A = \beta + \frac{2\beta\tilde{\ell}^2}{\tilde{\mu}^2} + \frac{4\beta\tilde{\ell}^2}{\tilde{\mu}^2(1 - \tilde{\ell}/\beta)^3}$, $\tilde{\mu} = \min\{\mu_1, \mu_2\}$.*

Based on this observation, we can therefore use the following easy-to-check criterion:

$$\|x - [x]_\beta\|^2 + \|y - [y]_\beta\|^2 \leq \min\left\{\frac{\tilde{\mu}^2 \epsilon^{(t)}}{2A(\beta + \tilde{\ell})^2}, \left(\frac{\tilde{\mu}\epsilon^{(t)}}{4\beta\mathcal{D}_{\mathcal{Y}}(\beta + \tilde{\ell})}\right)^2\right\}. \tag{6}$$

Note that many algorithms such as EG or GDA, already compute $([x]_\beta, [y]_\beta)$ with $\beta$ being the stepsize, so there is no additional computation cost to check criterion (6).

**Choice of regularization parameter.** As we can see, the smaller $\tau$ is, the auxiliary problem is closer to the original problem. However, smaller $\tau$ will give rise to worse conditions of the auxiliary problems, making them harder to solve. We will discuss the dependence of the inner and outer loop complexities on $\tau$ and provide a guideline for choosing $\tau$ for different $\mathcal{M}$.

As a final remark, we stress that the idea of using (accelerated) proximal point algorithm for minimax optimization is by no means new. Similar ideas have appeared in different contexts. However, they differ from our scheme in one way or the other. To list a few: [41, 30, 23, 38] considered the inexact PPA for C-C or NC-NC minimax problems by adding quadratic terms in both $x$ and $y$; [40, 44] considered the inexact PPA for NC-C minimax problems, by adding a quadratic term in $x$; [24] considered the inexact accelerated PPA for SC-SC minimax problems by adding a quadratic term in $x$. On the other hand, a number of work, e.g., [19, 24, 51] also add a quadratic term in $y$ to the minimax optimization, but in the form $\mathcal{O}(\epsilon)\|y\|^2$, which is completely different from PPA. Besides these differences, the subroutines used to solve the auxiliary minimax problems and choices of regularization parameters in these work are quite distinct from ours. Lastly, we point out that the proposed framework is closely related to the inexact accelerated augmented Lagrangian method designed for linearly constrained optimization problems [18], which can be viewed as a special case by setting $f(x, y)$ as the Lagrangian dual. In spite of this, approaches for solving the auxiliary problems are completely different, as is theoretical analysis.

## 3 Main Results

### 3.1 Convergence Analysis

In order to derive the total complexity, we first establish the complexity of the outer loop and then combine it with the inner loop complexity from algorithm $\mathcal{M}$. We then discuss the optimal choice of the regularization parameter $\tau$ for different settings.

**Theorem 3.1** (Outer-loop complexity). *Suppose function $f$ satisfies Assumptions 1 and 2. The output $(\bar{x}_T, y_T)$ from Algorithm 1 satisfies*

$$\operatorname{gap}_f(\bar{x}_T, y_T) \leq \alpha_T^2 \left[\frac{\tau}{2}\mathcal{D}_{\mathcal{Y}}^2 + 2\sum_{t=1}^T \frac{1}{\alpha_t^2}\epsilon^{(t)}\right], \tag{7}$$

*where $\mathcal{D}_{\mathcal{Y}} = \max_{y, y' \in \mathcal{Y}} \|y - y'\|$ is the diameter of $\mathcal{Y}$. If we further choose, $\epsilon^{(t)} = \frac{3\tau\mathcal{D}_{\mathcal{Y}}\alpha_t^2}{2\pi t^2}$, then*

$$\operatorname{gap}_f(\bar{x}_T, y_T) \leq \alpha_T^2 \tau \mathcal{D}_{\mathcal{Y}}^2. \tag{8}$$

**Remark 1.** *The above result is true without requiring strong convexity in $x$; only convexity-concavity of $f(x, y)$ is sufficient. In addition, the regularization parameter $\tau$ can be any positive value. Hence, Algorithm 1 is quite flexible. Because $2/(t + 2)^2 \leq \alpha_t^2 \leq 4/(t + 1)^2$ [39], Theorem 3.1 implies that the algorithm finds a point with $\epsilon$ primal-dual gap within $\mathcal{O}(\sqrt{\tau/\epsilon}\mathcal{D}_{\mathcal{Y}})$ outer-loop iterations. Notice that the outer-loop complexity decreases as $\tau$ decreases.*

We now turn to the inner loop complexity. By construction, the auxiliary problem $(\star)$ is $(\mu, \tau)$-SC-SC and $\tilde{\ell}$ smooth with $\tilde{\ell} = \ell + \tau$, which can be solved by many existing first-order algorithms at a linear convergence rate. Below we present the complexity of the inner loop with warm start.

**Proposition 3.1** (Inner-loop complexity). *Suppose we apply a linearly convergent algorithm $\mathcal{M}$ described by (4) or (5) to solve the auxiliary problem $(\star)$ and set the initial point to be $(x_{t-1}, z_t)$ at iteration $t$. Let $K(\epsilon^{(t)})$ denote the number of iterations (expected number of iterations if $\mathcal{M}$ is stochastic) for $\mathcal{M}$ to find a point satisfying (6). Then $K(\epsilon^{(t)})$ is $\mathcal{O}\left(\Delta_{\mathcal{M},\tau} \log\left(\frac{\tilde{\ell} \cdot \mathcal{D}_{\mathcal{Y}}}{\min\{1, \mu, \tau\} \cdot \epsilon^{(t)}}\right)\right)$.*

In practice, choosing a good initial point to warm start algorithm $\mathcal{M}$ can be helpful in accelerating the convergence. The above proposition shows that in theory, using a simple warm start strategy helps alleviate the logarithmic dependence on the distance from the initial point to the optimal point. Without the warm start strategy, one would require $\mathcal{X}$ to be bounded and $K(\epsilon^{(t)}) = \mathcal{O}\left(\Delta_{\mathcal{M},\tau} \log(\frac{\mathcal{D}_{\mathcal{X}} + \mathcal{D}_{\mathcal{Y}}}{\epsilon^{(t)}})\right)$. Here we do not require boundedness on $\mathcal{X}$.

As we can see, the choice of $\tau$ plays a crucial role since it affects both inner-loop and outer-loop complexities. Combining the above two results immediately leads to the total complexity:

**Corollary 3.2** (Total complexity). *Suppose Assumptions 1, 2 hold, and the subproblems are solved by a linearly convergent algorithm $\mathcal{M}$ to satisfy the stopping criterion (3) or (6) with accuracy $\epsilon^{(t)}$ as specified in Theorem 3.1. For Algorithm 1 to find an $\epsilon$-saddle point, the total number of gradient evaluations (expected number if $\mathcal{M}$ is stochastic) is*

$$\mathcal{O}\left(\Delta_{\mathcal{M},\tau}\sqrt{\tau/\epsilon}\mathcal{D}_{\mathcal{Y}} \log\left(\frac{\ell \cdot \mathcal{D}_{\mathcal{Y}}}{\min\{1, \mu, \tau\} \cdot \epsilon}\right)\right).$$

For any choice of linearly-convergent method $\mathcal{M}$ and any regularization parameter $\tau$, the oracle complexity is guaranteed to be $\mathcal{O}\left(\mathcal{D}_{\mathcal{Y}}/\sqrt{\epsilon} \log(\mathcal{D}_{\mathcal{Y}}/\epsilon)\right)$, which is optimal both in $\epsilon$ and $\mathcal{D}_{\mathcal{Y}}$ up to a logarithmic factor [37]. The dependence on the condition number will solely be determined by the term $\Delta_{\mathcal{M},\tau}\sqrt{\tau}$, which we analyze in detail below for specific algorithms.

## 3.2 Specific Algorithms and Complexities

In order to minimize the total complexity, we should choose the regularization parameter $\tau$ that $\min_{\tau > 0} \Delta_{\mathcal{M},\tau}\sqrt{\tau}$. Below we derive the choice of optimal $\tau$ for different algorithms $\mathcal{M}$ and present the corresponding total complexity. Table 2 summarizes this for several algorithms we consider.

**Deterministic first-order algorithms.** If we adopt the simplest gradient descent ascent (GDA) as $\mathcal{M}$ for solving the subproblem, then $\Delta_{\mathcal{M},\tau} = \left(\frac{\ell + \tau}{2\min\{\mu,\tau\}}\right)^2$ [12]. If $\mathcal{M}$ is extra-gradient method (EG) or optimistic gradient descent ascent (OGDA), then $\Delta_{\mathcal{M},\tau} = \frac{\ell + \tau}{4\min\{\mu,\tau\}}$ [45, 15, 2]. Minimizing $\Delta_{\mathcal{M},\tau}\sqrt{\tau}$ for both cases yields that the optimal choice for $\tau$ is $\mu$. In particular, when using EG or OGDA, the total complexity becomes

$$\mathcal{O}\left(\frac{\ell \cdot \mathcal{D}_{\mathcal{Y}}}{\sqrt{\mu\epsilon}} \log\left(\frac{\ell \cdot \mathcal{D}_{\mathcal{Y}}}{\min\{1, \mu\} \cdot \epsilon}\right)\right).$$

**Remark 2.** *This complexity matches the lower complexity bound for this class of problems [37] in $\epsilon, \ell, \mu$ and $\mathcal{D}_{\mathcal{Y}}$, up to a logarithmic factor. In addition, it improves over the best-known result, which was recently established in [24], which has a cubic order on the logarithmic factor and requires boundedness of $\mathcal{X}$.*

A key observation is that by setting $\tau = \mu$, the auxiliary problem $(\star)$ becomes $(\mu, \mu)$-SC-SC, and it is known that simple EG or OGDA achieves the optimal complexity for solving this class of well-balanced SC-SC problems [50]. Unlike [44, 24], their subproblems are harder to solve because of ill-balanced condition numbers, thus leading to an inferior complexity.

Besides the complexity improvement, our algorithm is significantly simpler and easier to implement than the current state-of-the-arts. The DIAG algorithm in [44] applies Nesterov's accelerated gradient ascent to the dual function and an additional two-loop algorithm to solve their subproblems. The MINIMAX-APPA algorithm in [24] adds a smoothing term in $y$ and applies a triple-loop algorithm to solve the auxiliary SC-SC problem. In contrast, our algorithm only requires two loops, does not require to prefix accuracy $\epsilon$, and has fewer tuning parameters. Results are summarized in Table 1.

Table 2: The table summarizes the optimal choice of regularization parameter $\tau$ and total complexity of the proposed Catalyst framework for finite-sum SC-C minimax optimization with $f(x,y) = \frac{1}{n}\sum_{i=1}^{n} f_i(x,y)$, when combined with different methods $\mathcal{M}$.

| $\mathcal{M}$ | $\Delta_{\mathcal{M},\tau} \propto$ | Choice for $\tau$ | Total Complexity of Catalyst |
|---|---|---|---|
| GDA [12] | $\left(\frac{\bar{\ell}+\tau}{\min\{\mu,\tau\}}\right)^2$ | $\mu$ | $\tilde{\mathcal{O}}\left(\frac{n\bar{\ell}^2}{\sqrt{\mu^3\epsilon}}\right)$ |
| EG/OGDA [29, 15] | $\frac{\bar{\ell}+\tau}{\min\{\mu,\tau\}}$ | $\mu$ | $\tilde{\mathcal{O}}\left(\frac{n\bar{\ell}}{\sqrt{\mu\epsilon}}\right)$ |
| SVRG/SAGA [38] | $n + \left(\frac{\bar{\ell}+\tau}{\min\{\mu,\tau\}}\right)^2$ | $\mu$, if $\bar{\ell}/\mu \geq \sqrt{n}$ $\frac{\bar{\ell}}{\sqrt{n}}$, if $\bar{\ell}/\mu < \sqrt{n}$ | $\tilde{\mathcal{O}}\left(\frac{\bar{\ell}^2}{\sqrt{\mu^3\epsilon}} \vee \frac{n^{\frac{3}{4}}\bar{\ell}^{\frac{1}{2}}}{\sqrt{\epsilon}}\right)$ |
| SVRE[1][6] | $n + \frac{\bar{\ell}+\tau}{\min\{\mu,\tau\}}$ | $\mu$, if $\bar{\ell}/\mu \geq n$ $\frac{\bar{\ell}}{n}$, if $\bar{\ell}/\mu < n$ | $\tilde{\mathcal{O}}\left(\frac{\bar{\ell}}{\sqrt{\mu\epsilon}} \vee \frac{n^{\frac{1}{2}}\bar{\ell}^{\frac{1}{2}}}{\sqrt{\epsilon}}\right)$ |

**Stochastic variance-reduced algorithms.** We now consider finite-sum-structure minimax problems, $\min_{x\in\mathcal{X}}\max_{y\in\mathcal{Y}} f(x,y) \triangleq \frac{1}{n}\sum_{i=1}^{n} f_i(x,y)$, where each component $f_i$ has $\ell_i$-Lipschitz gradients. Denote $\bar{\ell} = \frac{1}{n}\sum_{i=1}^{n}\ell_i$ as the average of smoothness constants. The resulting SC-SC subproblem $(\star)$ also has the finite-sum structure and can be solved by a number of linearly-convergent variance-reduced algorithms, such as SVRG, SAGA [38], and SVRE [6].

If using SVRG or SAGA as $\mathcal{M}$, we have $\Delta_{\mathcal{M},\tau} \propto n + \left(\frac{\bar{\ell}+\tau}{\min\{\mu,\tau\}}\right)^2$ [38]. When using SVRE as $\mathcal{M}$, $\Delta_{\mathcal{M},\tau} \propto n + \frac{\bar{\ell}+\tau}{\min\{\mu,\tau\}}$, assuming that the gradients are also $\ell_i$-cocoercive [6]. Particularly, when using SVRE, the optimal $\tau$ is $\mu$ if $\bar{\ell}/\mu \geq n$ and $\bar{\ell}/n$ otherwise. Therefore, the total complexity is

$$\tilde{\mathcal{O}}\left(\frac{\bar{\ell}}{\sqrt{\mu\epsilon}}\right) \text{ if } \bar{\ell}/\mu \geq n; \quad \text{and} \quad \tilde{\mathcal{O}}\left(\frac{n^{\frac{1}{2}}\bar{\ell}^{\frac{1}{2}}}{\sqrt{\epsilon}}\right) \text{ otherwise.}$$

**Remark 3.** *In either case, our result improves over the complexity $\tilde{\mathcal{O}}\left(\frac{n\bar{\ell}}{\sqrt{\mu\epsilon}}\right)$ when using the batch extra-gradient method as $\mathcal{M}$. To the best of our knowledge, this is the best complexity established so far for this class of SC-C minimax optimization problems. Results are summarized in Table 2.*

## 4 Nonconvex-Concave Minimax Optimization

We now turn to nonconvex-concave minimax problems (1), and formally make Assumption 3. Denote $g(x) = \max_{y\in\mathcal{Y}} f(x,y)$ as the primal function, which is $\ell$-weakly-convex [44]. The goal is to find an $\epsilon$-stationary point of $g(x)$. For any $\bar{x}$, consider the Moreau envelop of $g$: $\psi_{1/\tau_x}(\bar{x}) := \min_{x\in\mathcal{X}}\left\{g_{\tau_x}(x;\bar{x}) := g(x) + \frac{\tau_x}{2}\|x-\bar{x}\|^2\right\}$. The norm of the gradient $\|\nabla\psi_{1/\tau_x}(\bar{x})\|$ is commonly used to measure the quality of a solution $\bar{x}$ [10]. We call $\bar{x}$ $\epsilon$-stationary point of $g$ if $\|\nabla\psi_{1/\tau_x}(\bar{x})\| \leq \epsilon$.

**Assumption 3.** *$f(x,\cdot)$ is concave for any $x$ in $\mathcal{X}$. $\mathcal{X}$ and $\mathcal{Y}$ are convex and closed sets, and $\mathcal{Y}$ is bounded with diameter $\mathcal{D}_{\mathcal{Y}} = \max_{y,y'\in\mathcal{Y}}\|y-y'\|$.*

Our modified Catalyst framework is described in 2, which further applies the proximal point algorithm to the primal function $g(x)$, by adding a quadratic term in $x$, in the same spirit as [40, 44, 24]. The main difference lies in that we use Algorithm 1 to solve subproblems in form of $\min_{x\in\mathcal{X}} g_{\tau_x}(x;x_t)$. Now we use $\tau_y$ to denote the parameter in Algorithm 1 in order to distinguish from $\tau_x$. Algorithm 2 can be considered as a two-time-scale inexact proximal point algorithm, which repeatedly solves the subproblem

$$\min_{x\in\mathcal{X}}\max_{y\in\mathcal{Y}} f(x,y) + \frac{\tau_x}{2}\|x-\bar{x}_t\|^2 + \frac{\tau_y}{2}\|y-z_t\|^2. \tag{9}$$

We call it *two-time-scale*, not only because $\tau_x$ and $\tau_y$ differ, but also because the prox center of $y$ comes from the extrapolation step of acceleration and is updated more frequently than the prox center of $x$. The subproblem (9) is $(\tau_x - \ell, \tau_y)$-SC-SC if $\tau_x > \ell$, thus can be efficiently solved.

**Algorithm 2** `Catalyst for NC-C Minimax Optimization`

1: Input: initial point $(x_0, y_0)$, parameter $\tau_x > \ell$
2: **for all** $t = 0, 1, ..., T-1$ **do**
3:     use Algorithm 1 to find $x_{t+1}$ such that
$$g_{\tau_x}(x_{t+1}; x_t) \leq \min_{x \in \mathcal{X}} g_{\tau_x}(x; x_t) + \bar{\epsilon}$$
4: **end for**
5: Output: $\hat{x}_T$ which is uniformly sampled from $x_0, ..., x_{T-1}$.

---

**Theorem 4.1** (Outer-loop complexity). *Suppose $f$ satisfies Assumption* 2 *and* 3. *The output from Algorithm* 2 *satisfies*

$$\mathbb{E}\|\nabla\psi_{1/\tau_x}(\hat{x}_T)\|^2 \leq \frac{2\tau_x^2}{\tau_x - \ell}\left[\frac{g(x_0) - g^*}{T} + \bar{\epsilon}\right],$$

*where $g^* = \min_{x \in \mathcal{X}} g(x)$. If $T = \frac{4\tau_x^2(g(x_0) - g^*)}{(\tau_x - \ell)\epsilon^2}$ and $\bar{\epsilon} = \frac{(\tau_x - \ell)\epsilon^2}{2\tau_x^2}$, then $\mathbb{E}\|\nabla\psi_{1/\tau_x}(\hat{x}_T)\| \leq \epsilon$.*

Theorem 4.1 implies that the outer-loop complexity is $\mathcal{O}(\epsilon^{-2})$. In the following corollaries, we specify the choices of $\tau_x$, $\tau_y$, and $\mathcal{M}$ for solving subproblems and the total complexity.

**Corollary 4.2.** *Suppose $f$ satisfies Assumption* 2 *and* 3. *If we choose $\tau_x = 2\ell$, $\tau_y = \ell$ and use EG/OGDA/GDA to solve subproblems, then Algorithm* 2 *finds an $\epsilon$-stationary point with the total number of gradient evaluations of $\tilde{\mathcal{O}}\left(\ell^2\epsilon^{-3}\right)$.*

**Corollary 4.3.** *Suppose $f(x,y) = \frac{1}{n}\sum_{i=1}^n f_i(x,y)$ satisfies Assmption* 3 *and each component $f_i$ has $\ell_i$-Lipschitz gradient with $\bar{\ell} = \frac{1}{n}\sum_{i=1}^n \ell_i$. If we choose $\tau_x = 2\bar{\ell}$, $\tau_y = \frac{\bar{\ell}}{\sqrt{n}}$ and use SVRG/SAGA to solve subproblems, then Algorithm* 2 *finds an $\epsilon$-stationary point with the total complexity $\tilde{\mathcal{O}}\left(n^{\frac{3}{4}}\bar{\ell}^2\epsilon^{-3}\right)$. If we further assume $f_i$ has $\ell_i$-cocoercive gradient, choose $\tau_x = 2\bar{\ell}$, $\tau_y = \frac{\bar{\ell}}{n}$ and use SVRE to solve subproblems, then Algorithm* 2 *finds an $\epsilon$-stationary point with the total complexity $\tilde{\mathcal{O}}\left(n^{\frac{1}{2}}\bar{\ell}^2\epsilon^{-3}\right)$.*

Corollary 4.2 shows that simply using Catalyst-EG/OGDA achieves the complexity $\tilde{\mathcal{O}}\left(\ell^2\epsilon^{-3}\right)$. This matches with the current state-of-the-art complexity for nonconvex-concave minimization [24, 44, 51, 36]. Note that our algorithm is much simpler than the existing algorithms, e.g., Prox-DIAG [44] requires a four-loop procedure, whereas MINIMAX-APPA [24] requires a smoothing step. For problems with finite-sum structure, as shown in Corollary 4.3, using Catalyst-SVRG attains the overall complexity $\tilde{\mathcal{O}}\left(n^{\frac{3}{4}}\bar{\ell}^2\epsilon^{-3}\right)$, improving over all existing results. For instance, PG-SVRG proposed in [40] gives $\tilde{\mathcal{O}}\left(n\epsilon^{-2} + \epsilon^{-6}\right)$, which has a much worse dependence on $\epsilon$ and $n$.

## 5 Numerical Experiments

We consider the wireless communication problem in [3]. Given $n$ communications channels with signal power $p \in \mathbb{R}^n$ and noise power $\sigma \in \mathbb{R}^n$, the capacity of channel $i$ is proportional to $\log(1 + \beta_i p_i/(\sigma_i^0 + \sigma_i))$, where $\beta_i > 0$ and $\sigma_i^0$ are known constants. We would like to maximize the channel capacity under the adversarially chosen noise [14]. This can be formulated as an SC-C minimax problem:

$$\min_p \max_\sigma f(p, \sigma) := -\sum_{i=1}^n \log\left(1 + \frac{\beta_i p_i}{\sigma_i^0 + \sigma_i}\right) + \frac{\lambda}{2}\|p\|^2, \text{ such that } \mathbf{1}^\top\sigma = N, p \geq 0, \sigma \geq 0.$$

We generate two datasets with (1) $\beta = \mathbf{1}$ and $\sigma^0 \in \mathbb{R}^{1000}$ uniformly from $[0, 100]^{1000}$, (2) $\beta = \mathbf{1}$ and $\sigma^0 \in \mathbb{R}^{500}$ uniformly from $[0, 10]^{500}$. In Figure 1, we apply the same stepsizes to EG and subroutine in Catalyst-EG, and we compare their convergence results with stepsizes from small to large. In Figure 2, we compare four algorithms: extragradient (EG), SVRG, Catalyst-EG, Catalyst-SVRG with best-tuned stepsizes, and evaluate their errors based on (a) distance to the limit point: $\|p_t - p^*\| + \|\sigma_t - \sigma^*\|$; (b) norm of gradient mapping: $\|\nabla_p f(p_t, \sigma_t)\| + \|\sigma_t - \mathcal{P}_\Sigma(\sigma_t + \beta\nabla_\sigma f(p_t, \sigma_t))\|/\beta$. In Figure 3, we compare EG, Catalyst-EG and DIAG with best-tuned stepsizes.

Although EG with average iterates has an optimal complexity of $\mathcal{O}(1/\epsilon)$ for solving convex-concave minimax problems [32], its convergence behavior for SC-C minimax optimization remains unknown. Both Catalyst-EG and DIAG are designed for SC-C minimax optimization: Catalyst EG has a

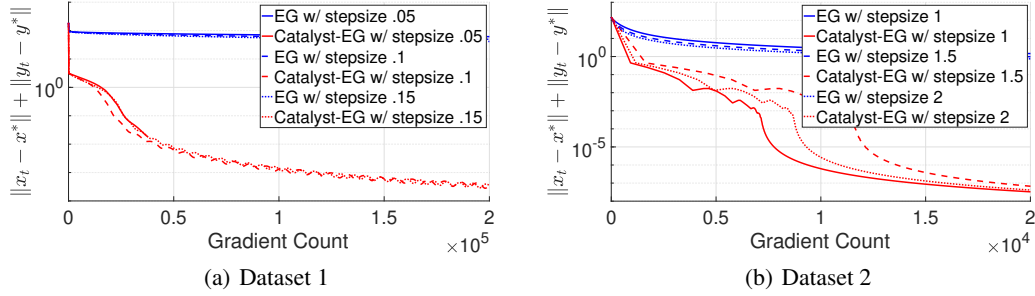

(a) Dataset 1

(b) Dataset 2

Figure 1: Comparion of EG and Caralyst-EG under same stepsizes

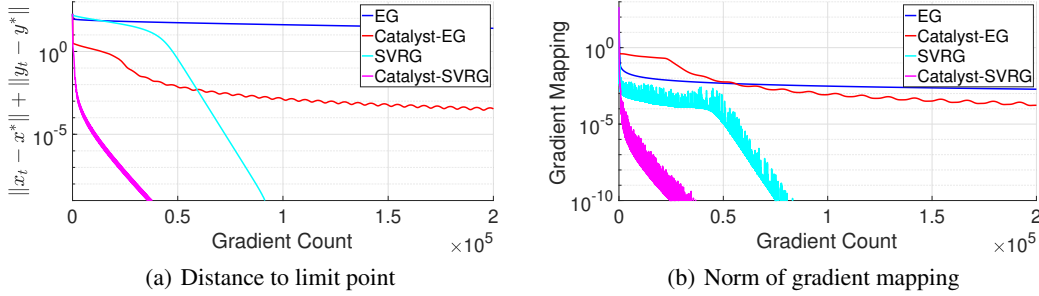

(a) Distance to limit point

(b) Norm of gradient mapping

Figure 2: Comparison of EG, Catalyst-EG, SVRG and Catalyst-SVRG on Dataset 1

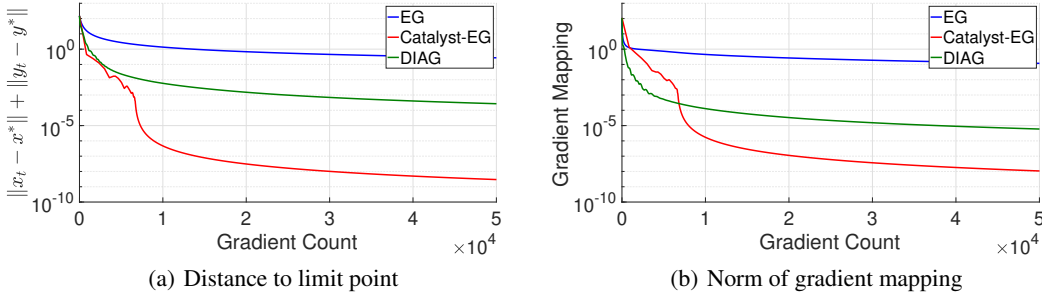

(a) Distance to limit point

(b) Norm of gradient mapping

Figure 3: Comparison of EG, Catalyst-EG, and DIAG on Dataset 2

complexity of $\tilde{\mathcal{O}}(\ell/\sqrt{\mu\epsilon})$ and DIAG has a complexity of $\tilde{\mathcal{O}}\left(\ell^{\frac{3}{2}}/(\mu\sqrt{\epsilon})\right)$. Here we use the same stepsize for primal and dual variables in EG and its counterpart with Catalyst. In Catalyst, we use $\|x_t - \mathcal{P}_{\mathcal{X}}(x_t - \beta\nabla_x f(x_t, y_t))\|/\beta + \|y_t - \mathcal{P}_{\mathcal{Y}}(y_t + \beta\nabla_y f(x_t, y_t))\|/\beta$ as stopping criterion for subproblem, which is discussed in Section 2. We control the subroutine accuracy $\epsilon^{(t)}$ as $\max\{c/t^8, \tilde{\epsilon}\}$, where $c$ is a constant and $\tilde{\epsilon}$ is a prefixed threshold. In contrast, DIAG does not provide a easy-to-verify stopping criterion for subroutine. We stop the subroutine of DIAG based on the criterion: $\|x_k - x_{k-1}\|^2 + \|y_k - y_{k-1}\|^2$, where $k$ indexes the subroutine iterations. We note that there is no theoretical convergence analysis for SVRG under SC-C setting. To form a fair comprison with SVRG, we report last iterate error in Catalyst-SVRG rather than averaged iterates.

We observe that Catalyst-EG performs better than EG and DIAG. Under the same stepsize, Catalyst framework significantly speed up EG. SVRG, albeit without theoretical guarantee in the SC-C setting, converges much faster than batch algorithms. Catalyst-SVRG also greatly improves over SVRG and outperforms all other algorithms.

## Acknowledgments and Disclosure of Funding

This work was supported in part by ONR grant W911NF-15-1-0479, NSF CCF-1704970, and NSF CMMI-1761699.

## Broader Impact

Our work provides a family of simple and efficient algorithms for some classes of minimax optimization. We believe our theoretical results advance many applications in ML which requires minimax optimization. Of particular interests are deep learning and fair machine learning.

Deep learning is used in many safety-critical environments, including self-driving car, biometric authentication, and so on. There is growing evidence that shows deep neural networks are vulnerable to adversarial attacks. Since adversarial attacks and defenses are often considered as two-player games, progress in minimax optimization will definitely empower both. Furthermore, minimax optimization problems provide insights and understanding into the balance and equilibrium between attacks and defenses. As a consequence, making good use of those techniques will boost the robustness of deep learning models and strengthen the security of its applications.

Fairness in machine learning has attracted much attention, because it is directly relevant to policy design and social welfare. For example, courts use COMPAS for recidivism prediction. Researchers have shown that bias is introduced into many machine learning systems through skewed data, limited features, etc. One approach to mitigate this is adding constraints into the system, which naturally gives rise to minimax problems.

## Footnotes

[1] SVRE requires assuming each component has $\ell_i$-cocoercive gradient, which is a stronger assumption than assuming $\ell_i$-Lipschitz gradient.

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
