[Reviews · NeurIPS 2020]

Review 1

Summary and Contributions: ****** After rebuttal ********************************** I am mostly satisfied by the answers of the authors, and wish to keep my score unchanged. In particular, would we allow a clear round of revisions, I would insist on adding: - at least a few numerical comparisons with smoothing techniques (which, imho, should be somehow presented, at least in appendix). - Complete comparisons/discussions of the results in terms of dependence on the diameters. - Cost of checking the stopping criterion for the stochastic case, for which implementation details matter (e.g., for SAGA). Therefore, the method is not totally generic and require at least a bit of thoughts about this (as an extreme case: paying n gradient evaluations per iteration would clearly not be acceptable). Beyond that, I wish to thank the authors for their answers! ******************************************************** The paper presents a "Catalyst" framework for strongly convex-concave minimax optimization, together with nearly optimal performance guarantees for this class of problems. In other words, the authors present an accelerated inexact proximal point method whose subproblems can be solved efficiently by any method that converge linearly on strongly convex-strongly concave problems. The main contribution is threefold: - improved worst-case complexity bounds for the class of strongly convex-concave problems - simplified ("two-loop") algorithm for doing it - using the strategy for developing a algorithm for nonconvex-concave minimax optimization.

Strengths: As far as I see, the contributions indeed seem new. - Theory seems OK (up to the typos) - The theory has the same flaws (e.g., "optimality up to the logarithmic factor") as similar works for optimization without saddle points, and therefore I believe (i) it can only slightly be improved, and (ii) it is more than reasonable to accept/trust. In addition, there is apparently no algorithm reaching the lower bound for this setting, making the results even nicer. - I believe the current interest for saddle point problems in ML makes such papers relevant for the community.

Weaknesses: On the other hand: - I believe more details should be provided for some parts of the proofs, at least in the appendix. I found a few typos, and therefore cannot exclude typos in the results themselves. - The approach critically requires knowledge of some of the problem's parameters (strong convexity, smoothness); I believe this should be discussed. - I believe some of the claims by the author(s) could be toned down (see detailed comments below). - The results themselves are probably not very surprising, nor very challenging, given the background results; although the idea is nice!

Correctness: I believe the theoretical claims are correct. However, comparisons with previous methods (in fact, only DIAG) are only presented in appendix: - I believe this should be presented in the core of the paper. - Do the authors see any way to compare to other methods, such as Minimax-APPA and smoothing? It would be fair to do it, although probably difficult due to the fact they require knowing the desired accuracy. However, for given access to the accuracy level (and some other problem parameters, such as distance to the solution), the comparison could be made.

Clarity: - The paper is nicely structured and mostly easy to read. - I would appreciate a revision of the appendix, where details are missing (although there is enough place for providing them).

Relation to Prior Work: - As far as I see, previous works are mostly well referred to. - I would probably add a reference to the original Catalyst paper "A Universal Catalyst for First-Order Optimization" by H Lin, J Mairal, and Z Harchaoui when referring to Catalyst.

Reproducibility: Yes

Additional Feedback: Questions in random ordering: - Would it be possible to provide dependences on the diameter(s) D_Y (and D_X?) in Table 1? - Reference for point (ii) page 3? - line 147: although this additional evaluation is certainly "negligible" for deterministic methods, is it really the case for stochastic ones? Was this cost taken into account in the numerical experiments? - Are there known results on using such Catalyst ideas on smooth convex-concave optimization? I guess there should be no gain (due to lower bound & EG), but e.g., do we also lose the logarithmic factor? - Is there a lower bound for the nonconvex-concave case? If not, please make it more explicit (e.g., in the abstract; "state-of-the-art" makes it a bit implicit) To go further: - Is it possible to use the method with raw estimates of mu and/or l? - (lines 42-54): Given that there is no known optimal algorithm; is it possible that the lower bound is not tight? (i.e., missing a log factor) Typos and minor comments: - imho, all sentences containing "[...] first [...]" should be toned down. In particular, in the abstract, the word "first" is probably a bit abusive, given that there exists closely related methods for closely related settings (e.g., [40]). In addition, it does not bring any information. Similar comment for line 87. - line 10: "[...] even faster convergence rate". Although I understand this sentence, such claims require an explicit comparison, which is not present. - line 13 (and later): "Catalyst framework"; imho, a reference to Lin, Mairal, and Harchaoui's paper is unavoidable when borrowing the same name. - line 19: "[...] often a smooth minimax [...]" I remain unsure about this claim: I believe it is much more often a nonsmooth minimax problem. What about "sometimes"? - There is a notation conflict between the "n" in the finite sum and the "n" being the dimension of y. - line 27: Miror -> Mirror - line 29: "limited" seems pretty harsh here, since it contains settings that are more general than that of the author(s). - in Section 2: there is a minor notation clash between z_t being the prox center for y's and for the concatenation of x's and y's (in e.g., Lemma 2.1). - Section 3.2: "[...] we should choose [...]". IMHO, this is not rigorously true: you make a (very reasonable) simplifying step. The rigorous optimization would take into account the "\tau" in the logarithm. - line 122: optimal gradient descent ascent -> optimistic GDA - Table 2: could the authors provide references in the table? - The appendix is quite not self-contained. For example, would it be possible to include a self-contained proof for (10)? Also later - Missing equation punctuation here and there. - Line 449: \tilde{f} -> \tilde{f}_t - line 455: \alpha -> \alpha_t - Some details for obtaining (28) would be nice. - Eq (29): missing square on the norm at the left hand side! - Eq 30: where do those "2" come from on the numerator and denominator? I guess from missing squares in (28) and (29)...


Review 2

Summary and Contributions: The present work proposes to solve strongly convex - weakly concave minimax problems by solving a series of proximal problems that are regularized with a quadratic term in the concave problem, resulting in strongly convex and strongly concave proximal problems. ****************************************************************************************** I thank the authors for addressing the my questions in their rebuttal. I will keep my score as-is since I am not fully convinced how impactful their method will be in practice but I believe that the authors have done all that can be expected in a NeurIPS paper to provide evidence for this claim. ****************************************************************************************** The inner loop strongly convex-concave problems can be solved efficiently using methods such as extragradient and allow for the simple incorporation of off-the shelf variance reduction methods such as the extragradient method.

Strengths: The authors propose a framework for using existing algorithms for strongly convex-concave minimax problems to solve problems that are only weakly concave. The availability of algorithms for these types of problems provides for considerable flexibility and both theoretical and empirical results support its superior convergence behavior.

Weaknesses: The first point of criticism is that the setting of strongly convex-weakly concave setting seems to be a narrow niche. This impression is fortified by the fact that the motivation given in the introduction advertises generic minimax problems. The authors mention that "In fact, strongly-convex-concave minimax optimization has been routinely used as a building block for solving nonconvex-concave minimax problems [43, 47]." but I believe this point should be made stronger to adquately motivate the paper. The second point of criticism pertains to the treatment of related work. My understanding is that competing methods are analyzed as using the same step size for both min and max player, whereas the proposed method has an additional hyperparameter (the regularization of the weakly concave maximizing player) that can induce behavior quite similar to the choice of seperate learning rate for min an max player. Therefore, it seems that a more fair comparison would allow for competing methods to pick seperate step sizes for the two players. Related to the above, it is unclear how the hyperparameters were choosen in the experiment. Finally, it seems like a comparison to recently proposed methods involving efficient hessian-vector products such as https://arxiv.org/pdf/1905.04926.pdf https://arxiv.org/abs/1808.01531 https://arxiv.org/abs/1709.04326 https://arxiv.org/abs/1905.12103 https://arxiv.org/abs/1705.10461 would be appropriate.

Correctness: As far as I can tell, this is the case

Clarity: The paper is well written

Relation to Prior Work: See "weaknesses" above.

Reproducibility: Yes

Additional Feedback: The idea of selectively regularizing the (only weakly convex) max player also appears in the classical method of augmented Lagrangians. Competitive gradient descent (https://arxiv.org/abs/1905.12103) is related to the work of the authors in that it uses a quadratically regularized bilinear approximation of the payoff function as a proximal problem.


Review 3

Summary and Contributions: After Rebuttal I am satisfied with the response from the authors as well as reading the other reviewers comments. I leave my score unchanged. %%%%%%%%%%%%%%%%%%%%%%% The authors extend the Catalyst framework of [Lin et al] to the smooth minimax optimization. Their contribution is threefold: (1). Strongly-convex-concave: By carefully choosing the parameter in the "proximal point" inner loop of Catalyst, the authors can improve all exciting methods including extra gradient yielding a complexity of O(ell/sqrt(mu epsilon) log(1/epsilon)) (2). Nonconvex-concave setting: the authors use Catalyst to accelerate existing methods to match the state-of-the-art complexity of O(ell^2/epsilon^(-3)) (3). Finite sum structure: The authors use Catalyst to construct the first variance reduction algorithm for strongly-convex-concave setting.

Strengths: The claims of the paper seem to the best of my knowledge accurate and theoretically grounded. Although I did not go through the details of the proofs, I am fairly confident that the results are true. Although using Catalyst is not new, the way the authors used Catalyst to solve minmax optimization problems is novel. Their approach is quite simple and easy to understand if one knows Catalyst already and it gives an easy way to accelerate known algorithms in the field. I also like that the algorithm does not require prior knowledge of the accuracy.

Weaknesses: (1). One big weakness with Catalyst in general is how to choose the parameters. In order for Catalyst to work, the knowledge of the ell and mu is essential. Have the authors considered a way to find these parameters so that the algorithm is more practical? I realize this is also a problem in Catalyst. (2). It is very similar to Catalyst and in that regard it isn't so surprising that this approach works. However, I do still like it. (3). One downside of NC-C setting is that you have to predefine the number of iterations to run the inner loop. Is there a way to remove this approach? This is very similar to predefining the accuracy and using this accuracy in the algorithm. (2).

Correctness: To the best of my knowledge, the claims appear to be correct. I did not check the proofs in detail but I am confident that they are correct.

Clarity: The paper is clearly written with just a few typos (e.g. Line 19).

Relation to Prior Work: The prior work is discussed in great detail. The authors appear knowledgeable in the area of minmax optimization as well as proximal point methods. They appear to have used the Catalyst algorithm in a novel way to accelerate methods which were not designed for stochastic optimization and applied it to minmax.

Reproducibility: Yes

Additional Feedback:


Review 4

Summary and Contributions: This paper extends a recently developed Catalyst framework for optimization. The original paper is Catalyst Acceleration for First-order Convex Optimization: from Theory to Practice, where the authors introduced the additional extrapolation step to the existing first order method to accelerate the asymptotic convergence behavior of existing optimization algorithms. This paper is an extension of this approach to a minimax problem.

Strengths: The extension is trivial in practice (which is good), while theoretical advancement is novel.

Weaknesses: The work is good, so I can't think of weakness of this work.

Correctness: I think the claim is correct, and the empirical methodology looks good. All the experiments supports theoretical claims

Clarity: The paper is well written.

Relation to Prior Work: Yes. The previous works are on the convex problem, non-convex problems but not for minimax problems which are more challenging to solve.

Reproducibility: Yes

Additional Feedback:

[Author Response · NeurIPS 2020]

We thank the reviewers for their careful reading and useful feedback. Below we briefly address each reviewer's
comments; these will be discussed in greater details in the revised version.

**Common question on how to choose the problem parameters:** As reviewers pointed out, the requirement to know
the problem parameters (strong convexity and smoothness) is endemic to many first order methods, including the
original Catalyst framework for minimization. It is worth pointing out that for many applications, the strong convexity
comes from the regularization term that is usually known. In practice, using *a rough estimate* of the strong convexity or
smoothness constant would also work as long as the choice of regularization parameter $\tau$ leads to a more balanced
auxiliary problem (which is one of the key insights of our framework). Such a rough estimate can be obtained by
approximately computing the spectrum of the Hessian or through backtracking. As an alternative, these constants can
be simply treated as hyper-parameters and tuned with the stepsize.

**To Reviewer 1:**

• **Numerical comparison:** Implementations of previous methods such as Minimax-APPA and primal-dual smoothing,
*are not available* in the original papers. These algorithms are difficult to implement as they require smoothing
and knowing the desired accuracy for each loop, which must be tuned in practice. That is why we pick DIAG as
a representative benchmark among these three-loop algorithms for numerical comparison. We will move these
comparisons from the appendix to the main paper and provide more numerical evidence.

• **Dependence on $\mathcal{D}_{\mathcal{Y}}$ and $\mathcal{D}_{\mathcal{X}}$:** All algorithms in Table 2 have the same dependency on $\mathcal{D}_{\mathcal{Y}}$. Minimax-APPA assumes
$\mathcal{X}$ to be bounded and has logarithmic dependency on $\mathcal{D}_{\mathcal{X}}$ in the complexity, and DIAG requires $\mathcal{X} = \mathbb{R}^d$. Our
algorithm assumes $\mathcal{X}$ is convex and only has logarithmic dependency on $\|x_0 - x^*\|$ in SC-C setting.

• **Negligible cost for checking stopping criterion**: For stochastic methods such as SVRG and SVRE, the cost for
checking the stopping criterion is $\mathcal{O}(n)$ in each epoch and does not increase the overall complexity. In addition, the
full gradient is already computed in each epoch, so the cost of checking this criterion is still almost negligible.

• **Gain in other setting.** As the reviewer pointed out, there is no gain for the convex-concave setting as EG is
already optimal. However, there could be gains in other settings, such as SC-SC and NC-SC settings. In fact, we
have recently shown that when extending to $(\mu_x, \mu_y)$-SC-SC setting, our framework achieves the complexity of
$\mathcal{O}(\ell/\sqrt{\mu_x \mu_y} \log(1/\epsilon))$, which again improves over EG and matches the lower bound up to logarithmic factors.

• **Lower bound**: No matching lower bound is known for the NC-C setting. A valid lower bound from nonconvex
minimization is $\mathcal{O}(1/\epsilon^2)$, which may not be tight. For the SC-C setting, we don not know if the current lower bound
is missing a log factor or not. We will explicitly discuss the open questions of lower bounds related to our settings.

• **Typos and appendix**: We will fix these issues and clean the appendix. *Thanks for the careful reading!*

**To Reviewer 2:**

• **Narrow niche**: We should have mentioned that strongly-convex-concave minimax optimization itself has broad
applications in game theory, imaging, distributionally robust optimization, etc. The bilinear case of this setting has
been studied extensively, but the general case remains unexplored. Besides, the framework we present for SC-C
setting can be extended to NC-C setting (as shown in the paper) and potentially other settings such as C-C, SC-SC,
and NC-SC settings (see our response to Review 1).

• **Different learning rates:** We agree with the reviewer that some related work uses different learning rates for the
two players for NC-C setting, e.g., GDA [Lin et al.,2019] uses $\mathcal{O}(\epsilon^4)$ stepsize for the min player and $\mathcal{O}(1)$ for the
max player. Theoretically, they achieve a much worse $\mathcal{O}(\epsilon^{-6})$ complexity than the $\mathcal{O}(\epsilon^{-3})$ complexity achieved by
our Catalyst with GDA under *the same constant stepsize* for both players. We will also add numerical comparisons.

• **Related work**: We thank the reviewer for the pointers. We will add these references and discuss them in the paper.
Competitive gradient descent [Schäfer & Anandkumar, 2019] and some recent work [Xu et al., 2020] add quadratic
regularization terms in the forms of $\|x\|^2$ and $\|y\|^2$. This is different from our framework, as we only add the term
$\|y - z_t\|^2$, where $z_t$ comes from extrapolation.

**To Reviewer 3:**
**Pre-defining the number of inner loop or accuracy**: This is a good point. In Theorem 4.1, we choose the desired
accuracy for inner loop to be $\bar{\epsilon} = \mathcal{O}(\epsilon^2)$. In fact, this can be replaced by an adaptive accuracy, i.e., $\bar{\epsilon}_t = \frac{g(x_0) - g^*}{t+1}$, which
will not increase the overall complexity. Hence, the number of inner loops does not necessarily have to be pre-defined.

**To Reviewer 4:**
We thank the reviewer for the acknowledgement of our contribution. Although Catalyst framework in minimization
provide a strong intuition for our work, the analysis for the outer-loop complexity of our framework is an nontrivial
extension of their analysis as it requires evaluating solution performance in terms of primal-dual gap (which is much
stronger than dual optimality).

[Meta-Review · NeurIPS 2020]

The paper received positive feedback. After reading the rebuttal and discussing the paper, the general consensus is that the paper should be accepted. The area chair agrees with this assessement and follows the reviewer's recommendation. Several suggestions were made to improve the paper (see in particular R1's review), which will be good to take into account for the final version.